# Parsing-based Approaches for Verification and Recognition of Hierarchical Plans

**Roman Barták**
Charles University
Faculty of Mathematics and Physics
Prague, Czech Republic

**Adrien Maillard**
GMV
Flight Dynamics and Operations
Toulouse, France

**Rafael C. Cardoso**
University of Liverpool
Department of Computer Science
Liverpool, United Kingdom

## Abstract

Hierarchical planning, in particular, Hierarchical Task Networks, was proposed as a method to describe plans by decomposition of tasks to sub-tasks until primitive tasks, actions, are obtained. Plan verification assumes a complete plan as input, and the objective is finding a task that decomposes to this plan. In plan recognition, a prefix of the plan is given and the objective is finding a task that decomposes to the (shortest) plan with the given prefix. This paper describes how to verify and recognize plans using a common method known from formal grammars, by parsing.

## Introduction

Hierarchical planning is a practically important approach to automated planning based on encoding abstract plans as *hierarchical task networks* (HTNs) (Erol, Hendler, and Nau 1996). The network describes how compound tasks are decomposed, via decomposition methods, to sub-tasks and eventually to actions forming a plan. The decomposition methods may specify additional constraints among the sub-tasks such as partial ordering and causal links.

As of this writing, there exist only two systems for *verifying* if a given plan complies with the HTN model, that is, if a given sequence of actions can be obtained by decomposing some task. One system is based on transforming the verification problem to SAT (Behnke, Höller, and Biundo 2017) and the other system is using parsing of attribute grammars (Barták, Maillard, and Cardoso 2018). Only the parsing-based system supports HTN fully (the SAT-based system does not support the decomposition constraints).

Parsing became popular in solving the *plan recognition problem* (Vilain 1990) as researchers realized soon the similarity between hierarchical plans and formal grammars, specifically context-free grammars with parsing trees close to decomposition trees of HTNs. The plan recognition problem can be formulated as the problem of adding a sequence of actions after some observed partial plan such that the joint sequence of actions forms a complete plan generated from some task (more general formulations also exist). Hence plan recognition can be seen as a generalization of plan verification. There exist numerous approaches to plan recognition using parsing or string rewriting (Avrahami-Zilberbrand

and Kaminka 2005; Geib, Maraist, and Goldman 2008; Geib and Goldman 2009; Kabanza et al. 2013), but they use hierarchical models that are weaker than HTNs. The languages defined by HTN planning problems (with partial-order, preconditions and effects) lie somewhere between context-free (CF) and context-sensitive (CS) languages (Höller et al. 2014) so to model HTNs, one needs to go beyond the CF grammars. Currently, the only grammar-based model that fully covers HTNs uses attribute grammars (Barták and Maillard 2017). Moreover, the expressivity of HTNs makes the plan recognition problem undecidable (Behnke, Höller, and Biundo 2015). At the moment, there is only one approach for HTN plan recognition. This approach relies on translating the plan recognition problem to a planning problem (Höller et al. 2018), which is a technique that was first introduced in (Ramírez and Geffner 2003).

In this paper, we focus on verification and recognition of HTN plans using parsing. The uniqueness of the proposed methods is that they cover full HTNs including task interleaving, partial order of sub-tasks, and other decomposition constraints (prevailing constraints, specifically). The methods are derived from the plan verification technique proposed in (Barták, Maillard, and Cardoso 2018).

There are two novel contributions of this paper. First, we will simplify the above mentioned verification technique by exploiting information about actions and states to improve practical efficiency of plan verification. Second, we will extend that technique to solve the plan (task) recognition problem. We will show that the verification algorithm can be much simpler and, hence, it is expected to be more efficient. For plan recognition, the method proposed in (Höller et al. 2018) can in principle support HTN fully, if a full HTN planner is used (which is not the case yet as prevailing conditions are not supported). However, like other plan recognition techniques it requires the top task (the goal) and the initial state to be specified as input. A practical difference of our methods is that they do not require information about possible top (root) tasks and an initial state as their input. This is particularly interesting for plan/task recognition, where existing methods require a set of candidate tasks (goals) to select from (in principle, they may use all tasks as candidates, but this makes them inefficient).

# Background on Planning

In this paper, we work with classical STRIPS planning (Fikes and Nilsson 1971) that deals with sequences of actions transferring the world from a given initial state to a state satisfying certain goal conditions. World states are modelled as sets of propositions that are true in those states, and actions are are modelled to change the validity of certain propositions.

## Classical Planning

Formally, let $P$ be a set of all propositions modelling properties of world states. Then a state $S \subseteq P$ is a set of propositions that are true in that state (every other proposition is false). Later, we will use the notation $S^+ = S$ to describe explicitly the valid propositions in the state $S$ and $S^- = P \setminus S$ to describe explicitly the propositions not valid in the state $S$.

Each action $a$ is described by three sets of propositions $(B_a^+, A_a^+, A_a^-)$, where $B_a^+, A_a^+, A_a^- \subseteq P, A_a^+ \cap A_a^- = \emptyset$. Set $B_a^+$ describes positive preconditions of action $a$, that is, propositions that must be true right before the action $a$. Some modeling approaches allow also negative preconditions, but these preconditions can be compiled away. For simplicity reasons we assume positive preconditions only (the techniques presented in this paper can also be extended to cover negative preconditions directly). Action $a$ is applicable to state $S$ iff $B_a^+ \subseteq S$. Sets $A_a^+$ and $A_a^-$ describe positive and negative effects of action $a$, that is, propositions that will become true and false in the state right after executing the action $a$. If an action $a$ is applicable to state $S$ then the state right after the action $a$ is:

$$\gamma(S, a) = (S \setminus A_a^-) \cup A_a^+. \tag{1}$$

$\gamma(S, a)$ is undefined if an action $a$ is not applicable to state $S$.

The classical planning problem, also called a STRIPS problem, consists of a set of actions $A$, a set of propositions $S_0$ called an initial state, and a set of goal propositions $G^+$ describing the propositions required to be true in the goal state (again, negative goal is not assumed as it can be compiled away). A solution to the planning problem is a sequence of actions $a_1, a_2, \ldots, a_n$ such that $S = \gamma(\ldots\gamma(\gamma(S_0, a_1), a_2), \ldots, a_n)$ and $G^+ \subseteq S$. This sequence of actions is called a *plan*.

The *plan verification problem* is formulated as follows: given a sequence of actions $a_1, a_2, \ldots, a_n$, and goal propositions $G^+$, is there an initial state $S_0$ such that the sequence of actions forms a valid plan leading from $S_0$ to a goal state? In some formulations, the initial state might also be given as an input to the verification problem.

## Hierarchical Task Networks

To simplify and speed up the planning process, several extensions of the basic STRIPS model were proposed to include some control knowledge. Hierarchical Task Networks (Erol, Hendler, and Nau 1996) were proposed as a planning domain modeling framework that includes control knowledge in the form of recipes on how to solve specific tasks.

The recipe is represented as a task network, which is a set of sub-tasks to solve a given task together with the set of constraints between the sub-tasks. Let $T$ be a compound task and $(\{T_1, ..., T_k\}, C)$ be a task network, where $C$ are its constraints (see later). We can describe the decomposition method as a derivation (rewriting) rule:

$$T \to T_1, ..., T_k \ \ [C]$$

The planning problem in HTN is specified by an initial state (the set of propositions that hold at the beginning) and by an initial task representing the goal. The compound tasks need to be decomposed via decomposition methods until a set of primitive tasks – actions – is obtained. Moreover, these actions need to be linearly ordered to satisfy all the constraints obtained during decompositions and the obtained plan – a linear sequence of actions – must be applicable to the initial state in the same sense as in classical planning. We denote an action as $a_i$, where the index $i$ means the order number of the action in the plan ($a_i$ is the $i$-th action in the plan). The state right after the action $a_i$ is denoted $S_i$, $S_0$ is the initial state. We denote the set of actions to which a task $T$ decomposes as $act(T)$. If $U$ is a set of tasks, we define $act(U) = \cup_{T \in U} act(T)$. The index of the first action in the decomposition of $T$ is denoted $start(T)$, that is, $start(T) = min\{i|a_i \in act(T)\}$. Similarly, $end(T)$ means the index of the last action in the decomposition of $T$, that is, $end(T) = max\{i|a_i \in act(T)\}$.

We can now define formally the constraints $C$ used in the decomposition methods. The constraints can be of the following three types:

- $t_1 \prec t_2$: a precedence constraint meaning that in every plan the last action obtained from task $t_1$ is before the first action obtained from task $t_2$, $end(t_1) < start(t_2)$,

- $before(U, p)$: a precondition constraint meaning that in every plan the proposition $p$ holds in the state right before the first action obtained from tasks $U$, $p \in S_{start(U)-1}$,

- $between(U, V, p)$: a prevailing condition meaning that in every plan the proposition $p$ holds in all the states between the last action obtained from tasks $U$ and the first action obtained from tasks $V$,
$\forall i \in \{end(U), \ldots, start(V) - 1\}, p \in S_i$.

The *HTN plan verification problem* is formulated as follows: given a sequence of actions $a_1, a_2, \ldots, a_n$, is there an initial state $S_0$ such that the sequence of actions is a valid plan applicable to $S_0$ and obtained from some compound task? Again, the initial state might also be given as an input in some formulations.

The *HTN plan recognition problem* is formulated as follows: given a sequence of actions $a_1, a_2, \ldots, a_n$, is there an initial state $S_0$ and actions $a_{n+1}, \ldots, a_{n+m}$ for some $m \geq 0$ such that the sequence of actions $a_1, a_2, \ldots, a_{n+m}$ is a valid plan applicable to $S_0$ and obtained from some compound task? In other words, if the given actions form a prefix of some plan obtained from some compound task $T$. We will be looking for such a task $T$ minimizing the value $m$ (the number of added actions to complete the plan). If only the task $T$ is of interest (not the actions $a_{n+1}, \ldots, a_{n+m}$) then it can be referred to as the *task (goal) recognition problem*.

## The Plan Verification Algorithm

The existing parsing-based HTN verification algorithm (Barták, Maillard, and Cardoso 2018) uses a complex structure of a timeline. This structure maintains the decomposition constraints so that they can be checked when composing sub-tasks to a compound task. We propose a simplified verification method that does not require this complex structure, as it checks all the constraints directly in the input plan. This makes the algorithm easier to implement and also potentially faster. Another difference is that we do not assume that the initial state is passed as input, instead we set the initial state as the preconditions of the first action in the plan. However, adding support for it is trivial as we would only have to add the initial state that was given as input to the preconditions of the first action in the plan.

The novel hierarchical plan verification algorithm is shown in Algorithm 1. It first calculates all intermediate states (lines 2-8) by propagating information about propositions in action preconditions and effects. At this stage, we actually solve the classical plan validation problem as the algorithm verifies that the given plan is causally consistent (action precondition is provided by previous actions or by the initial state). The original verification algorithm did this calculation repeatedly each time it composed a compound task. It is easy to show that every action is applicable, that is, $B_{a_i}^+ \subseteq S_{i-1}$ (lines 2 and 4). Next, we will show that $\gamma(S_i, a_{i+1}) = S_{i+1} = (S_i \setminus A_{a_{i+1}}^-) \cup A_{a_{i+1}}^+$. Left-to-right propagation (line 4) ensures that $(S_i \setminus A_{a_{i+1}}^-) \cup A_{a_{i+1}}^+ \subseteq S_{i+1}$. Right-to-left propagation (line 6) ensures that preconditions are propagated to earlier states if not provided by the action at a given position. In other words, if there is a proposition $p \in S_{i+1} \setminus A_{a_{i+1}}^+$ then this proposition should be at $S_i$. Line 6 adds such propositions to $S_i$ so it holds $(S_i \setminus A_{a_{i+1}}^-) \cup A_{a_{i+1}}^+ = S_{i+1}$. However, if $p \in A_{a_{i+1}}^-$ then $p$ would be deleted by the action $a_{i+1}$, which means that the plan is not valid. The algorithm detects this at lines 7-8.

When the states are calculated, we apply a parsing algorithm to compose tasks. Parsing starts with the set of primitive tasks (line 9), each corresponding to an action from the input plan. For each task $T$, we keep a data structure describing the set $act(T)$, that is, the set of actions to which the task decomposes. We use a Boolean vector $I$ of the same size as the plan to describe this set; $a_i \in act(T) \Leftrightarrow I(i) = 1$. To simplify checks of decomposition constraints, we also keep information about the index of first and last actions from $act(T)$. Together, the task is represented using a quadruplet $(T, b, e, I)$ in which $T$ is a task, $b$ is the index in the plan of the first action generated by $T$, $e$ is the index in the plan of the last action generated by $T$ (we say that $[i, j]$ represents the interval of actions over which $T$ *spans*), and $I$ is a Boolean vector as described above.

The algorithm applies each decomposition rule to compose a new task from already known sub-tasks (line 12). The composition consists of merging the sub-tasks, when we check that every action in the decomposition is obtained from a single sub-task (line 20), that is, $act(T_0) = \bigcup_{j=1}^k act(T_j)$ and $\forall i \neq j : act(T_i) \cap act(T_j) = \emptyset$. We also check all the decomposition constraints; the pseudo-code is

**Data:** a plan $\mathbf{P} = (a_1, ..., a_n)$ and a set of decomp. methods

**Result:** a Boolean equal to true if the plan can be derived from some compound task, false otherwise

1 **Function** VERIFYPLAN
2   $S_0 \leftarrow B_{a_1}^+$
3   **for** $i = 1$ **to** $n$ **do**
4    $S_i \leftarrow B_{a_{i+1}}^+ \cup (S_{i-1} \setminus A_{a_i}^-) \cup A_{a_i}^+$
5   **for** $i = n\text{-}1$ **down to** $0$ **do**
6    $S_i \leftarrow S_i \cup (S_{i+1} \setminus A_{a_{i+1}}^+)$
7    **if** $S_i \cap A_{a_i}^- \neq \emptyset$ **then**
8     **return false**
9   $\mathbf{sp} \leftarrow \emptyset; \text{new} \leftarrow \{(A_i, i, i, I_i) \mid i \in 1..n\}$
  **Data:** $A_i$ is a primitive task corresponding to action $a_i$, $I_i$ is a Boolean vector of size $n$, such that $\forall i \in 1..n, I_i(i) = 1, \forall j \neq i, I_i(j) = 0$
10   **while** $\text{new} \neq \emptyset$ **do**
11    $\mathbf{sp} \leftarrow \mathbf{sp} \cup \text{new}; \text{new} \leftarrow \emptyset$
12    **foreach** *decomposition method $R$ of the form* $T_0 \rightarrow T_1, ..., T_k$ $[\prec, \text{pre}, \text{btw}]$ *such that* $\{(T_j, b_j, e_j, I_j) \mid j \in 1..k\} \subseteq \mathbf{sp}$ **do**
13     **if** $\exists (i, j) \in \prec : \neg(e_i < b_j)$ **then**
14      **break**
15     $b_0 \leftarrow \min\{b_j \mid j \in 1..k\}$
16     $e_0 \leftarrow \max\{e_j \mid j \in 1..k\}$
17     **for** $i = 1$ **to** $n$ **do**
18      $I_0(i) \leftarrow \sum_{j=1}^k I_j(i);$
19      **if** $I_0(i) > 1$ **then**
20       **break**
21     **if** $\exists (U, p) \in \text{pre} : p \notin S_{\min\{b_j \mid j \in U\}-1}$ **then**
22      **break**
23     **if** $\exists (U, V, p) \in \text{btw}\ \exists i \in \max\{e_j \mid j \in U\}, \ldots, \min\{b_j \mid j \in V\} - 1 : p \notin S_i$ **then**
24      **break**
25     $\text{new} \leftarrow \text{new} \cup \{(T_0, b_0, e_0, I_0)\}$
26     **if** $\forall k, I_0(k) = 1$ **then**
27      **return true**
28   **return false**

**Algorithm 1:** Parsing-based HTN plan verification

a direct rewriting of constraint definitions. If all tests pass, the new task is added to a set of tasks (line 25). Then we know that the task decomposes to actions, which form a sub-sequence (not necessarily continuous) of the plan to be verified. The process is repeated until a task that decomposes to all actions is obtained (line 27) or no new task can be composed (line 10). The algorithm is *sound* as the returned task decomposes to all actions in the input plan. If the algorithm finishes with the value **false** then no other task can be derived. As there is a finite number of possible tasks, the algorithm has to finish, so it is *complete*.

## The Plan Recognition Algorithm

Any plan verification algorithm, for example, the one from the previous section, can be extended to plan recognition by feeding the verification algorithm with actions $a_1, \ldots, a_{n+k}$, where we progressively increase $k$. The actions $a_1, \ldots, a_n$ are given as an input, while the actions $a_{n+1}, \ldots, a_{n+k}$ need to be generated (planned). However, this generate-and-verify approach would be inefficient for larger $k$ as it requires exploration of all valid sequences of actions with the prefix $a_1, \ldots, a_n$. Assume that there could be 5 actions at the position $n+1$ and 6 actions at the position $n+2$. Then the generate-and-verify approach explores up to 30 plans (not every action at the position $n+2$ could follow every action at the position $n+1$) and for each plan the verification part starts from scratch as the plans are different.

This is where the verification algorithm from (Barták, Maillard, and Cardoso 2018) can be used as it does not require exactly one action at each position. The algorithm stores actions (sub-tasks) independently and only when it combines them to form a new task, it generates the states between the actions and checks the constraints for them. This resembles the idea of the Graphplan algorithm (Blum and Furst 1997). There are also sets of candidate actions for each position in the plan and the plan-extraction stage of the algorithm selects some of them to form a causally valid plan. We use compound tasks together with their decomposition constraints to select and combine the actions (we do not use parallel actions in the plan).

The algorithm from (Barták, Maillard, and Cardoso 2018) extended to the plan recognition problem is shown in Algorithm 2. It starts with actions $a_1, \ldots, a_n$ (line 2) and it finds all compound tasks that decompose to subsets of these actions (lines 4-30). This inner while-loop is taken from (Barták, Maillard, and Cardoso 2018), we only syntactically modified it to highlight the similarity with the verification algorithm from the previous section. If a task that decomposes to all current actions is found (line 30) then we are done. This is the goal task that we looked for and its timeline describes the recognized plan. Otherwise, we add all primitive tasks corresponding to possible actions at position $n+1$ (line 33). Note that these are not parallel actions, the algorithm selects exactly one of them for the plan.

Now, the parsing algorithm continues as it may compose new tasks that include one of those recently added primitive tasks. Notice that the algorithm uses all composed tasks from previous iterations in succeeding iterations so it does not start from scratch when new actions are added. This process is repeated until the goal task is found. The algorithm is clearly *sound* as the task found is the task that decomposes to the shortest plan with a given prefix. This goes from the soundness and completeness of the verification algorithm (in particular, no task that decomposes to a shorter plan exists). The algorithm is *semi-complete* as if there exists a plan with the length $n+k$ and with a given prefix, the algorithm will eventually find it at the $(k+1)$-th iteration. If no plan with a given prefix exists then the algorithm will not stop. However, recall that the plan recognition problem is undecidable (Behnke, Höller, and Biundo 2015) so any plan recognition approach suffers from this deficiency.

**Data:** a plan $\mathbf{P} = (a_1, ..., a_n)$, $A_i$ is a primitive task corresponding to action $a_i$, and a set of decomposition methods

**Result:** a Task that decomposes to a plan with prefix $\mathbf{P}$

**1 Function** RECOGNIZEPLAN

**2** $\quad$ new $\leftarrow \{(A_i, i, i, \{(B_{a_i}^+, \emptyset, a_i, A_{a_i}^+, A_{a_i}^-)_i\}) | i \in 1..n\}$ ;

**3** $\quad$ $\mathbf{sp} \leftarrow \emptyset; l \leftarrow n$;

**4** $\quad$ **while** new $\neq \emptyset$ **do**

**5** $\quad\quad$ $\mathbf{sp} \leftarrow \mathbf{sp} \cup$ new; new $\leftarrow \emptyset$;

**6** $\quad\quad$ **foreach** *decomposition method $R$ of the form* $T_0 \to T_1, ..., T_k[\prec, \mathrm{pre}, \mathrm{btw}]$ *such that* $\{(T_j, b_j, e_j, tl_j) | j \in 1..k\} \subseteq \mathbf{sp}$ **do**

**7** $\quad\quad\quad$ **if** $\exists (i,j) \in \prec : \neg (e_i < b_j)$ **then**

**8** $\quad\quad\quad\quad$ **break**

**9** $\quad\quad\quad$ $b_0 \leftarrow \min\{b_j | j \in 1..k\}$

**10** $\quad\quad\quad$ $e_0 \leftarrow \max\{e_j | j \in 1..k\}$

**11** $\quad\quad\quad$ $tl \leftarrow \{(\emptyset, \emptyset, empty, \emptyset, \emptyset)_i | i \in b_0..e_0\}$

**12** $\quad\quad\quad$ **for** $j = 1$ **to** $k$; $i = b_j$ **to** $e_j$ **do**

**13** $\quad\quad\quad\quad$ $(\mathrm{Pre}_1^+, \mathrm{Pre}_1^-, a_1, \mathrm{Post}_1^+, \mathrm{Post}_1^-)_i \in tl$

**14** $\quad\quad\quad\quad$ $(\mathrm{Pre}_2^+, \mathrm{Pre}_2^-, a_2, \mathrm{Post}_2^+, \mathrm{Post}_2^-)_i \in tl_j$

**15** $\quad\quad\quad\quad$ **if** $a_1 \neq empty, a_2 \neq empty$ **then**

**16** $\quad\quad\quad\quad\quad$ **break**

**17** $\quad\quad\quad\quad$ $\mathrm{Pre}_1^+ \leftarrow \mathrm{Pre}_1^+ \cup \mathrm{Pre}_2^+$

**18** $\quad\quad\quad\quad$ $\mathrm{Pre}_1^- \leftarrow \mathrm{Pre}_1^- \cup \mathrm{Pre}_2^-$

**19** $\quad\quad\quad\quad$ $\mathrm{Post}_1^+ \leftarrow \mathrm{Post}_1^+ \cup \mathrm{Post}_2^+$

**20** $\quad\quad\quad\quad$ $\mathrm{Post}_1^- \leftarrow \mathrm{Post}_1^- \cup \mathrm{Post}_2^-$

**21** $\quad\quad\quad\quad$ **if** $a_1 = empty$ **then**

**22** $\quad\quad\quad\quad\quad$ $a_1 \leftarrow a_2$

**23** $\quad\quad\quad$ APPLYPRE$(tl, pre)$;

**24** $\quad\quad\quad$ APPLYBETWEEN$(tl, btw)$;

**25** $\quad\quad\quad$ PROPAGATE$(tl, b_0, e_0 - 1)$;

**26** $\quad\quad\quad$ **if** $\exists (\mathrm{Pre}^+, \mathrm{Pre}^-, a, \mathrm{Post}^+, \mathrm{Post}^-) \in tl :$ $\mathrm{Pre}^+ \cap \mathrm{Pre}^- \neq \emptyset$ **then**

**27** $\quad\quad\quad\quad$ **break**

**28** $\quad\quad\quad$ new $\leftarrow$ new $\cup \{(T_0, b_0, e_0, tl)\}$

**29** $\quad\quad\quad$ **if** $b_0 = 1, e_0 = l, \forall (\_, \_, a_j, \_, \_)_j \in tl :$ $a_j \neq empty$ **then**

**30** $\quad\quad\quad\quad$ **return** $(T_0, tl)$

**31** $\quad$ $l \leftarrow l + 1$;

**32** $\quad$ new $\leftarrow \{(A, l, l, \{(B_a^+, \emptyset, a, A_a^+, A_a^-)_l\}) |$

**33** $\quad$ action $a$ can be at position $l$; $A$ is a primitive task for $a\}$

**34** $\quad$ **goto** 4

**Algorithm 2:** Parsing-based HTN plan recognition

The algorithm maintains a timeline for each compound task to verify all the constraints. This is the major difference from the above verification algorithm that points to the original plan. This timeline has been introduced in (Barták, Maillard, and Cardoso 2018), where all technical details can be found. We include a short description to make the paper self-contained. A *timeline* is an ordered sequence of slots, where each slot describes an action, its effects, and the state right

before the action. For task $T$, the actions in slots are exactly the actions from $act(T)$. Both effects and states are modelled using two sets of propositions, $\text{Post}^+$ and $\text{Post}^-$ modeling positive and negative effects of the action and $\text{Pre}^+$ and $\text{Pre}^-$ modeling propositions that must and must not be the true in the state right before the action. Two sets are used as the state is specified only partially and propositions are added to it during propagation so it is necessary to keep information about propositions that must not be true in the state.

The timeline always spans from the first to the last action of the task. Due to interleaving of tasks (actions from one task might be located between the actions of another task in the plan), some slots of the task might be *empty*. These empty slots describe "space" for actions of other tasks. When we are merging sub-tasks (lines 12-22), we merge their timelines, slot by slot. This is how the actions from sub-tasks are put together in a compound task. Notice, specifically, that it is not allowed for two merged sub-tasks to have actions in the same slot (line 15). This ensures that each action is generated by exactly one task.

---

**Data:** a set of $slots$, a set of $before$ constraints
**Result:** an updated set of slots
1 **Function** APPLYPRE($slots, pre$)
2    **foreach** $(U, l) \in pre$ **do**
3       $id = \min\{b_j | j \in U\}$;
4       $\text{Pre}_{id}^+ \leftarrow \text{Pre}_{id}^+ \cup \{p | l = p\}$;
5       $\text{Pre}_{id}^- \leftarrow \text{Pre}_{id}^- \cup \{p | l = \neg p\}$

**Algorithm 3:** Apply before constraints

---

**Data:** a set of $slots$, a set of $between$ constraints
**Result:** an updated set of slots
1 **Function** APPLYBETWEEN($slots, between$)
2    **foreach** $(U, V, l) \in between$ **do**
3       $s = \max\{e_i | i \in U\} + 1$;
4       $e = \min\{b_i | i \in V\}$;
5       **for** $id = s$ **to** $e$ **do**
6          $\text{Pre}_{id}^+ \leftarrow \text{Pre}_{id}^+ \cup \{p | l = p\}$;
7          $\text{Pre}_{id}^- \leftarrow \text{Pre}_{id}^- \cup \{p | l = \neg p\}$

**Algorithm 4:** Apply between constraints

---

Propositions from $before$ and $between$ constraints are "stored" in the corresponding slots (Algorithms 3 and 4) and their consistency is checked each time the slots are modified (line 26 of Algorithm 2). Consistency means that no proposition is true and false at the same state. Information between subsequent slots is propagated similarly to the verification algorithm (see Algorithm 5). Positive and negative propositions are now propagated separately taking in account empty slots. If there is no action in the slot then effects are unknown and hence propositions cannot be propagated.

---

**Data:** a set of slots $slots$
**Result:** an updated set of slots
1 **Function** PROPAGATE($slots, lb, ub$)
   /\* Propagation to the right   \*/
2    **for** $i = lb$ **to** $ub$ **do**
3       **if** $a_i \neq empty$ **then**
4          $\text{Pre}_{i+1}^+ \leftarrow$
            $\text{Pre}_{i+1}^+ \cup (\text{Pre}_i^+ \setminus \text{Post}_i^-) \cup \text{Post}_i^+$;
5          $\text{Pre}_{i+1}^- \leftarrow$
            $\text{Pre}_{i+1}^- \cup (\text{Pre}_i^- \setminus \text{Post}_i^+) \cup \text{Post}_i^-$
   /\* Propagation to the left   \*/
6    **for** $i = ub$ **down to** $lb$ **do**
7       **if** $a_i \neq empty$ **then**
8          $\text{Pre}_i^+ \leftarrow \text{Pre}_i^+ \cup (\text{Pre}_{i+1}^+ \setminus \text{Post}_i^+)$;
9          $\text{Pre}_i^- \leftarrow \text{Pre}_i^- \cup (\text{Pre}_{i+1}^- \setminus \text{Post}_i^-)$

**Algorithm 5:** Propagate

---

## Example

A unique property of the proposed techniques is handling task interleaving – actions generated from different tasks may interleave to form a plan. This is the property that parsing techniques based on CF grammars cannot handle.

The example in Figure 1 demonstrates how the timelines are filled by actions as the tasks are being derived/composed from the plan. Assume, first, that a complete plan consisting of actions $a_1, a_2, \ldots, a_7$ is given. The plan recognition algorithm can also handle such situations, when a complete plan is given, so it can serve for plan verification too (the verification variant of Algorithm 2 should stop with a failure at line 33 as no action can be added during plan verification). In the first iteration, the algorithm will compose tasks $T_2, T_3, T_4$ as these tasks decompose to actions directly. Notice, how the timelines with empty slots are constructed. We know where the empty slots are located as we know the exact location of actions in the plan. In the second iteration, only the task $T_1$ is composed from already known tasks $T_3$ and $T_4$. Notice how the slots from these tasks are copied to the slots of a new timeline for $T_1$. On the contrary, the slots in original tasks remain untouched as these tasks may merge with other tasks to form alternative decomposition trees (see the discussion below). Finally, in the third iteration, tasks $T_1$ and $T_2$ are merged to a new task $T_0$ and the algorithm stops there as a complete timeline that fully spans the plan is obtained (condition at line 30 of Algorithm 2 is satisfied).

Let us assume that there is a constraint $between(\{a_1\}, \{a_3\}, p)$ in the decomposition method for $T_3$. For example, this constraint may model a causal link between $a_1$ and $a_3$. When composing the task $T_3$, the second slot of its timeline remains empty, but the proposition $p$ is placed there (see Algorithm 4). This proposition is then copied to the timeline of task $T_1$, when merging the timelines (line 17 of Algorithm 2), and finally also to the timeline of task $T_0$. During each merge operation, the algorithm checks that $p$ can still be in the slot, in particular, that $p$ is not required to be false at the same slot

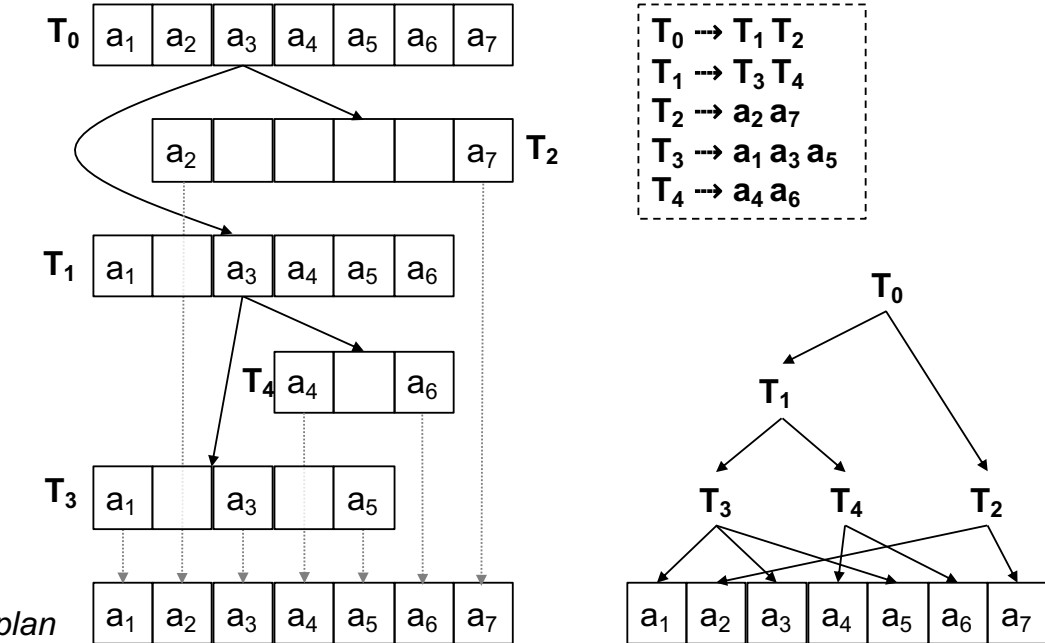

Figure 1: Example of parsing-based plan verification/recognition (the right side shows the decomposition tree with the decomposition rules above it; the left side shows the tasks with timelines and filled slots)

(line 26). Algorithm 2 repeatedly checks the constraints from methods.

The new plan verification algorithm (Algorithm 1) handles the method constraints more efficiently as it uses the complete plan with states to check them. Moreover, the propagation of states is run just once in Algorithm 1 (lines 2-8), while Algorithm 2 runs it repeatedly each time the task is composed from subtasks. Hence, each constraint is verified just once in Algorithm 1, when a new task is composed. In particular, the constraint $between(\{a_1\}, \{a_3\}, p)$ is verified with respect to the states when task $T_3$ is introduced. Otherwise, both Algorithm 1 and Algorithm 2 derive the tasks in the same order (if the decomposition methods are explored in the same order). Instead of timelines, Algorithm 1 uses the Boolean vector $I$ to identify actions belonging to each task. For example, for task $T_3$ the vector is $[1, 0, 1, 0, 1, 0, 0]$ and for task $T_4$ it is $[0, 0, 0, 1, 0, 1, 0]$. When composing task $T_1$ from $T_3$ and $T_4$ the vectors are merged to get $[1, 0, 1, 1, 1, 1, 0]$ (see the loop at line 17). Notice that the vector always spans the whole plan, while the timelines start at the first action and finish with the last action of the task (and hence the same timeline can be used for different plan lengths).

Assume now that only plan prefix consisting of $a_1, a_2, \ldots, a_6$ is given. The plan recognition algorithm (Algorithm 2) will first derive tasks $T_3$ and $T_4$ only. Specifically, task $T_2$ cannot be derived yet as action $a_7$ is not in the plan. In the second iteration, the algorithm will derive task $T_1$ by merging tasks $T_3$ and $T_4$, exactly as we described above. As no more tasks can be derived, the inner loop finishes and the algorithm attempts to add actions that can follow the prefix

$a_1, a_2, \ldots, a_6$ (line 33). Let action $a_7$ be added at the 7-th position in the plan; actually all actions, that can follow the prefix, will be added as separate primitive tasks at position 7. Now the inner loop is restarted and task $T_2$ will be added in its first iteration. In the next iteration, task $T_0$ will be added and this will be the final task as it satisfies the condition at line 30.

Assume, hypothetically, that the verification Algorithm 1 is used there. When it is applied to plan $a_1, a_2, \ldots, a_6$, the algorithm derives tasks $T_1, T_3, T_4$ and fails as no task spans the whole plan and no more tasks can be derived. After adding action $a_7$, the algorithm will start from scratch as the states might be different due to propagating some propositions from the precondition of $a_7$. Hence, the algorithm needs to derive the tasks $T_1, T_3, T_4$ again and it will also add tasks $T_0, T_2$ and then it will finish with success.

It may happen, that action $a_5$ can also be consistently placed to position 7. Then, we can derive two versions of task $T_3$, one with the vector $[1, 0, 1, 0, 1, 0, 0]$ and the other one with vector $[1, 0, 1, 0, 0, 0, 1]$. Let us denote the second version as $T_3'$. Both versions can then be merged with task $T_4$ to get two versions of task $T_1$, one with the vector $[1, 0, 1, 1, 1, 1, 0]$ and one with the vector $[1, 0, 1, 1, 0, 1, 1]$. Let us denote the second version as $T_1'$. The Algorithm 1 will stop there as no more tasks can be derived. Notice that tasks $T_1, T_3, T_4$ were derived repeatedly. If we try $a_5$ earlier than $a_7$ at position 7 then tasks $T_1, T_3, T_4$ will actually be generated three times before the algorithm finds a complete plan. On the contrary, Algorithm 2 will add actions $a_5$ and $a_7$ together as two possible primitive tasks at position 7. It will use tasks $T_1, T_3, T_4$ from the previous iteration, it will

add tasks $T_1', T_3'$ as they can be composed from the primitive tasks (using the last $a_5$), it will also add tasks $T_0, T_2$ (using the last $a_7$), and will finish with success. Notice that $T_1'$ cannot be merged with $T_2$ to get a new $T_0'$ as $T_1'$ has action $a_5$ at the 7-th slot while $T_2$ has $a_7$ there so the timelines cannot be merged (line 15 of Algorithm 2).

## Possible Extensions

To describe the verification and recognition algorithms, we used a "pure" model of HTN. Specifically, each task decomposes to a non-empty set of sub-tasks, meaning that the right-hand side of each derivation rule is non-empty. In some practical applications, it might be useful to also use decomposition methods with empty task networks. Imagine a task describing that some agent moves to a specific location. This task can be full-filled by action *move* so there will be a method, where the task decomposes to this action. However, if the agent is already at the specific location then no action is necessary and the task is already full-filled. This can be modeled by an alternative method that decomposes the task to an empty task network with the precondition (*before*) constraint specifying that the agent is at the required location. Such empty methods can be compiled away, for example, using the techniques for converting grammars to a normal form. Nevertheless, the presented verification and recognition algorithms can also be extended to handle derivation rules with empty right-hand side. We will demonstrate this extension for the verification Algorithm 1. Note, that tasks that decompose to an empty task network are treated in a similar way as tasks that decompose directly to actions, that is, they are added in the initialization stage (line 9). We only need to identify the proper location indexes of these tasks and this is where the $before$ constraint can be used. Assume the following method with empty right-hand side:

$$T \to \emptyset \ \ [before(\emptyset, p)].$$

First, the constraint $before(U, p)$ has originally been defined for a non-empty subset $U$ of tasks in the task network, but the task network is now empty so, in this special case, we allow $U = \emptyset$. Second, the verification algorithm already calculated all the states $S_i$ between the actions. The precondition constraint tells us, where the task $T$ can be inserted. Specifically, if $p \in S_i$, that is, the precondition constraint holds at state $S_i$, then we add a primitive task $(T, i + 0.1, i + 0.1, I)$ to the initial set of tasks *new* (line 9 of Algorithm 1), where the Boolean vector $I$ consists of zeros only (the task $T$ does not decompose to any action). We use the $(i + 0.1)$ index as the task $T$ is sitting between actions $A_i$ and $A_{i+1}$ and we need to ensure that possible precedence constraints involving $T$ work fine. The rest of the verification algorithm remains without further modification, we only need to properly round the indexes when checking the state constraints.

The second extension, that we are going to discuss, is about the top task to be recognized/verified. Recall, that neither of the proposed techniques requires a top task to be given at input. In some applications, a task network with constraints is given as input and the plan should correspond to this network. This can be trivially handled by the

proposed techniques by introducing, to the HTN model, a dummy root task that decomposes to this task network and modifying the terminal conditions of the algorithms to look for this specific root task rather than for any task (line 27 of Algorithm 1 and line 30 of Algorithm 2). However, what if the plan consists of interleaved sub-plans obtained from several tasks that are not known a priori? This situation can also be handled by modifying the termination condition. Instead of looking for a single task that spans the whole plan, we need to look for a set of already recognized tasks such that they do not share any action and, together, they span the whole plan. Note, however, that such a test can be computationally expensive if implemented in a naive way by checking all subsets of tasks.

## Conclusions

In the paper, we proposed two versions of a parsing technique for verification of HTN plans and for recognition of HTN plans. To the best of our knowledge, these are the only approaches that fully cover HTN, including all decomposition constraints. These approaches can be applied to solve both verification and recognition problems, but as we demonstrated using an example, each of them has some deficiencies when applied to the other problem.

The next obvious step is implementation and empirical evaluation of both techniques. There is no doubt that the novel verification algorithm is faster than the previous approaches (Behnke, Höller, and Biundo 2017) and (Barták, Maillard, and Cardoso 2018). The open question is how much faster it will be, in particular for large plans. The efficiency of the novel plan recognition technique in comparison to existing compilation technique (Höller et al. 2018) is less clear as both techniques use different approaches, bottom-up vs. top-down. The disadvantage of the compilation technique is that it needs to re-generate the known plan prefix, but it can exploit heuristics to remove some overhead there. On the contrary, the parsing techniques looks more like generate-and-test, but controlled by the hierarchical structure. It also guarantees finding the shortest extension of plan prefix.

**Acknowledgements**   Research is supported by the Czech Science Foundation under the project P103-18-07252S.

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
