# OpenReview forum: "Parsing-based Approaches for Verification and Recognition of Hierarchical Plans"
_icaps-conference.org/ICAPS/2019/Workshop/Hierarchical_Planning_

### Official Review · AnonReviewer1 · 2019-04-18
**Good paper. Promising work. Longing to see experiment results.**

**Rating:** 8
**Confidence:** 4
**Overall Evaluation:** 1
**Significance:** 2
**Soundness:** 3
**Clarity:** 2

**Review:**

This paper is about plan verification and recognition of HTN plans using parsing. This topic is very relevant for this workshop. Though presenting preliminary works, this paper introduces novel contributions: it enhances and extends previous techniques. Bibliography to my knowledge seems ok. There is little to say except that the writing is sometime really hard to follow due to over formalisation. In my opinion, ideas should be put forward rather than equations. In "The plan verification algorithm", formulas/algorithms don't say it all. To enhance the paper, I would use the "example" section as a running and/or introductory example. Moreover, evaluating this work is difficult as no experiment results and comparisons are provided. However, I share author views on probable performance improvements.

Questions :
- Your technique seems limited to single task goal. Can you give some hints on how to extend it to set of tasks + constraints?
- It is not clear to me how to translate concrete planning domains into (T, s, e, I). Can you be more specific on that?

Little typo in the introduction : "A practical difference of OUT methods" -> OUR ?
"By contrary/Contrary" -> On the contrary ?
The meaning of i line 13 in algo. 1 is not clear

**Reproducibility:**

3: authors describe the implementation and domains in sufficient detail

**Reviewer'S Confidence:**

3: medium

**Scholarship:**

2: relevant literature cited but could be expanded

---

### Official Review · AnonReviewer3 · 2019-04-25
**High quality approach to plan recognition for HTN**

**Rating:** 9
**Confidence:** 4
**Overall Evaluation:** 3
**Significance:** 3
**Soundness:** 3
**Clarity:** 3

**Review:**

The paper provides parsing approaches for HTN plan verification/recognition that are reminiscent of the dynamic programming proofs used by Erol et al. (94) and Alford et al. (2014) for various HTN complexity results.  I am quite interested in whether the empirical success of the Barták 2018 verification algorithm can be extended to plan recognition, especially in early stages of a plan when the remainder of a complete plan may be significantly longer than the completed portion.

There are two potential extensions that occur to me: One is breaking down Alg 2 into heuristic search over candidate completions, and the other is whether you can use ongoing work on either optimal HTN heuristics or  bounding HTN solutions for SAT translation to prune certain candidate solution lengths.

The paper is well written; clearly situating the problem in the literature, describing the algorithms, and running through an example.  The long term impact of the work is dependent on a thorough empirical evaluation.  Hopefully the workshop can provide discussion and ideas that are valuable in that effort.

[Errata]
Pg 1. "out methods"

**Reproducibility:**

2: some details missing but still appears to be replicable with some effort

**Reviewer'S Confidence:**

4: high

**Scholarship:**

3: excellent coverage of related work